# Chromosome-level assembly of the common vetch *(Vicia sativa)* reference genome

Hangwei Xi[1], Vy Nguyen[1], Christopher Ward[1], Zhipeng Liu[2,*] and Iain R. Searle[1,*]

1 School of Biological Sciences, The University of Adelaide, Adelaide, Adelaide 5005, Australia
2 State Key Laboratory of Grassland Agro-ecosystems, College of Pastoral Agriculture Science and Technology, Lanzhou University, No 768 Jiayuguan West Road, Chengguan District, Lanzhou 730020, China

## ABSTRACT

*Vicia sativa* L. (common vetch, $n$ = 6) is an annual, herbaceous, climbing legume, originating in the Fertile Crescent of the Middle East and now widespread in the Mediterranean basin, West, Central and Eastern Asia, North and South America. *V. sativa* is of economic importance as a forage legume in countries such as Australia, China, and the USA, and contributes valuable nitrogen to agricultural rotation cropping systems. To accelerate precision genome breeding and genomics-based selection of this legume, we present a chromosome-level reference genome sequence for *V. sativa*, constructed using a combination of long-read Oxford Nanopore sequencing, short-read Illumina sequencing, and high-throughput chromosome conformation data (CHiCAGO and Hi-C) analysis. The chromosome-level assembly of six pseudo-chromosomes has a total genome length of 1.65 Gbp, with a median contig length of 684 Kbp. BUSCO analysis of the assembly demonstrated very high completeness of 98% of the dicotyledonous orthologs. RNA-seq analysis and gene modelling enabled the annotation of 53,218 protein-coding genes. This *V. sativa* assembly will provide insights into vetch genome evolution and be a valuable resource for genomic breeding, genetic diversity and for understanding adaption to diverse arid environments.

**Subjects** Genetics and Genomics, Bioinformatics, Plant Genetics

**Submitted:** 17 October 2021

\* Corresponding authors. E-mail: lzp@lzu.edu.cn; iain.searle@adelaide.edu.au

Preprint submitted at https://doi.org/10.1101/2021.10.11.464017

## DATA DESCRIPTION

### Background

*Vicia sativa* L. (common vetch, NCBI:txid3908) (Figure 1) is an annual legume belonging to the Fabaceae family, and *Vicia* genus [1]. The *Vicia* genus contains about 180–210 species, including the economically important crop broad bean [2]. To date, no chromosome-level genome assembly has been reported within the *Vicia* genus. Interestingly, *V. sativa* has at least three different reported haploid chromosome numbers: $n$ = 5, 6 or 7 [3], but $n$ = 6 is the best characterized karyotype.

V. sativa is thought to have originated in the Fertile Crescent of the Middle East and is now widespread on every continent as both a crop and a weed [4]. *V. sativa* is a multipurpose legume; the plants are often grown for forage and the seeds can be used safely as a feed for ruminant animals. *V. sativa* seed contains up to 30% crude protein and is rich in essential amino acids and unsaturated fatty acids [5]. However, only a small amount of the seed can be safely fed to monogastric animals like chickens and pigs, because of the

**Figure 1.** *Vicia sativa* (cultivar Studenica). (A) Ten-week-old *V. sativa* at flowering. (B) Detached stem showing compound leaves at each node. At the end of each compound leaf is a tendril. A single pod forms at the base of each leaf node after flowering (arrow). (C) A shoot apex with a flower and surrounding young leaves. (D) Young to mature seed pods (left to right), with a representative seed shown at the bottom of each pod. Each pod contains 3–5 seeds. (E) Dry seeds of cultivar Studenica. Scale bars (A) = 10 cm, (B, C, E) = 1 cm, (D) = 2 cm.

presence of the neurotoxic proteinaceous amino acids $\beta$-cyano-L-alanine and $\gamma$-glutamyl-$\beta$-cyano-alanine [6].

*V. sativa* is often used in crop rotation systems to increase nitrogen input to the soil. In a study of *V. sativa*/wheat rotation over a 4-year-period, cultivation of *V. sativa* during autumn increased soil water storage and subsequently increased biological yield and grain yield of wheat. Both yields were doubled in the third year compared with the second year of the rotation [7]. Furthermore, the symbiosis between soil rhizobia bacteria and *V. sativa*

roots allows the plant to fix atmospheric nitrogen and later provide nitrogen for the following crop, hence reducing the use of expensive nitrogen fertilizer [8]. *V. sativa* exhibits excellent drought tolerance and is suitable for cultivation in arid areas. In one drought tolerance study, *V. sativa* could withstand a month of drought stress, with the leaf weight not decreasing significantly compared with the non-drought control [9]. *V. sativa* offers multiple usage and is a valuable crop in a sustainable agricultural system [10].

With the important value of *V. sativa*, vetch breeders have primarily selected for traits conferring high yield, pod shattering, flowering time, disease resistance against *Ascochyta fabae*, *Uromyces viciae-fabae* (rust) and *Sclerotinia sclerotium* [11]. Recently published transcriptome data has allowed agriculturally important traits to be uncovered at the gene expression level, such as pod-shattering resistance [12] and drought tolerance genes [13] in *V. sativa*. However, a lack of high-quality genome reference is currently impeding the genetic mapping of important genes and hindering further applications such as genome editing when compared with other crops.

## Context

In this study, we assembled a high-quality chromosome-level reference genome for *V. sativa*, which is the first chromosome-level reference genome in the *Vicia* genus. We performed genome annotation using RNA-seq data from five tissues to ensure most of the expressed genes were captured. We also included a phylogenetic analysis of *V. sativa* and legume relatives. We envisage that our *V. sativa* genome will be an important resource for evolutionary studies of this species. The well-annotated chromosome-level genome will also provide important information to facilitate genetic mapping, gene discovery and functional gene studies.

## METHODS

## Sampling and sequencing

To prepare *V. sativa* for whole genome sequencing (WGS) using long-read and short-read data, seeds of cultivar Studenica (*V. sativa* subsp. *sativa*) were obtained from the South Australian Research and Development Institute (SARDI, South Australia, Australia). Seeds were sterilized and germinated *in vitro* on half-strength Murashige & Skoog (1/2 MS) basal medium with 1% sucrose for 3 days at 25 °C, in the dark. Bulk 3-mm-long primary root tips were then harvested and snap-frozen in liquid nitrogen for subsequent DNA extraction. DNA was extracted using the phenol:chloroform method [14], with an additional high-salt low-ethanol wash to improve DNA purity [15]. High-quality DNA was confirmed by electrophoresis on 1% agarose gel. The DNA was sent to the Australian Genome Research Facility (AGRF, Melbourne, Australia), and Novogene Co., Ltd (Hong Kong, China) for library preparation and sequencing on a PromethION (PromethION, RRID:SCR_017987) and Novo-Seq 6000 (Illumina NovaSeq 6000 Sequencing System, RRID:SCR_016387), respectively. We obtained 72 gigabase pairs (Gbp) of Nanopore long-read data, and 205 Gb paired-end short-read data (150 base pairs [bp] read length).

To produce *V. sativa* CHiCAGO sequencing data [16] and Hi-C sequencing data [17], 2 g of young leaf tissue was snap-frozen in liquid nitrogen and sent to Dovetail Genomics (USA) for library preparation and sequencing. CHiCAGO and Hi-C libraries were sequenced on an Illumina HiSeq X (Illumina HiSeq X Ten, RRID:SCR_016385) to produce 162 Gbp of CHiCAGO and 148 Gbp of Hi-C sequencing data, respectively.

**Table 1.** Overview of sequencing data generated in this study.

| Libraries | Insert size (bp) | Raw data (Gbp) | Clean data (Gbp) | Mean read length (bp) | Coverage (×)* |
|---|---|---|---|---|---|
| WGS Illumina short-reads | 300 | 205.13 | 200.28 | 150 | 124.32 |
| Nanopore reads | N/A | 72.12 | N/A | 9094 | 43.71 |
| CHiCAGO | 350 | 162.00 | N/A | 150 | 98.18 |
| Hi-C | 350 | 147.60 | N/A | 150 | 89.45 |
| Illumina RNA-seq reads | 300 | 74.60 | 66.49 | 150 | 45.21 |

*Coverage was calculated based on the assembled genome size of 1.65 Gbp.

To prepare *V. sativa* RNA sequencing (RNA-seq) data, RNA was purified from the first two fully expanded leaves, shoot apexes with young leaves up to 1 cm long from 4-week-old plants, roots from 5-day-old seedlings and 4-week-old leaf-derived callus tissues using the Spectrum™ Plant Total RNA Kit (Sigma Aldrich). Additional DNase I treatment was used to remove DNA contamination (On-Column DNase I Digestion, Sigma Aldrich), and ribosome removal treatment to enrich for the non-ribosomal RNA fraction (Ribo-Zero rRNA Removal Kit for Plant Leaf or Plant Seed/Root, Illumina) [18]. Directional RNA libraries were prepared for each tissue using the NEBNext Ultra™ Directional RNA Library Prep Kit for Illumina (New England Biolabs) following the manufacturer's protocol. Libraries were sent to Novogene Co., Ltd (Hong Kong, China) for sequencing on Novo-Seq 6000 (Illumina) to obtain 150-bp paired-end read data. In total, we obtained 74.6 Gbp of RNA-seq data. A summary of the long and short-read sequencing data is provided in Table 1.

## Genome size estimation and genome assembly

We first performed a genome size estimation for *V. sativa*. To do this, short-Illumina (paired-end 150 nt) reads were trimmed using TrimGalore v0.4.2 (Trim Galore, RRID:SCR_011847) with default parameters and 25-mers were counted using Jellyfish v2.2.6 (Jellyfish, RRID:SCR_005491) [19]. The 25-mer count distribution data was used as an input to GenomeScope (GenomeScope, RRID:SCR_017014) [20] for genome size estimation with the read length set to 150 and max *k*-mer coverage set to 1 million. GenomeScope estimated a genome-wide heterozygosity level of 0.09% (Figure 2) and a genome size of 1.61 Gbp; approximately 160 megabase pairs (Mbp) smaller than the genome size estimated by flow cytometry (1.77 Gbp) [21].

Next, we conducted contig assembly from the Nanopore long-reads using Canu v1.7 (Canu, RRID:SCR_015880) [22] under default parameters with the expected genome size set at 1.77 Gbp. Canu was used to perform read trimming and sequencing error correction for the input data before performing contig assembly. The assembled contigs were polished using clean WGS short-reads with Pilon v1.22 (Pilon, RRID:SCR_014731) [23] under default parameters. We repeated the polishing step and observed a further improvement in contig quality (Table 2). Contig quality was assessed using BUSCO v5.2.2 (BUSCO, RRID:SCR_015008) [24] for the completeness of the genome, and after two rounds of polishing, complete BUSCOs increased from 69.9% to 97.8% (Table 2). Overall, we obtained 9,990 assembled contigs, which were 1.93 Gbp, with an N50 value of 685 kilobase pairs (Kbp).

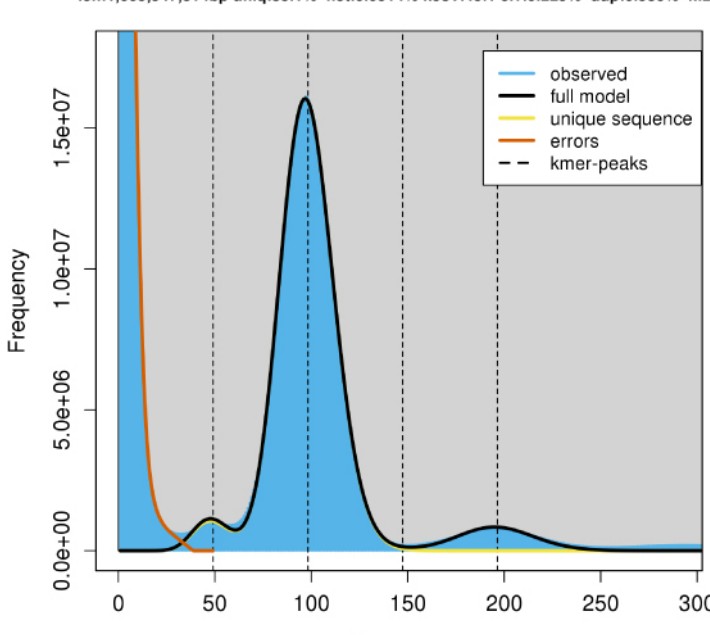

**Figure 2.** 25-mer distribution of Illumina paired-end reads by using GenomeScope. 25-mer occurrences (*x* axis) were plotted against their frequencies (*y* axis). Estimated genome size: 1,609,547,814 bp, estimated unique region: 35.1%, estimated heterozygosity: 0.0914%, estimated genome.

**Table 2.** Genome completeness evaluated by BUSCO with fabales_odb10 dataset (number of BUSCOs: 5366) after the first and second round of polishing *V. sativa* contigs using WGS short-read data.

| BUSCO analysis | No polishing (%) | 1st polishing (%) | 2nd polishing (%) |
|---|---|---|---|
| Complete | 69.9 | 97.7 | 97.8 |
| Complete and single-copy | 63 | 87.3 | 88.9 |
| Complete and duplicated | 6.9 | 10.4 | 8.9 |
| Fragmented | 3.5 | 0.3 | 0.3 |
| Missing | 26.6 | 2.0 | 1.9 |

## Chromosome-level assembly using Hi-C and linkage map data

To generate a chromosome-level assembly for *V. sativa*, Hi-C proximity [25] ligation data and CHiCAGO [26] were used to order and orient the contigs along chromosomes. The scaffolding process was carried out by Dovetail Genomics (Santa Cruz, CA, USA) using Dovetail™ Hi-C library reads to connect and order the input set of contigs. After scaffolding with HiRise (v2.1.7) [51], the assembled genome sequence initially comprised 1.8 Gbp, with a scaffold and contig N50 of 51.4 and 0.6 Mbp, respectively. A high fraction of the assembled sequences (93%) were contained in four pseudo-chromosomes (Figure 3A); however *V. sativa* has six pairs of chromosomes [1]. We observed that two of the four pseudo-chromosomes had weak interactions, suggesting misjoining of two contigs (Figure 3A).

In parallel to the HiRise analysis, we performed a second chromosome-level assembly using 3D-DNA (3D de novo assembly, RRID:SCR_017227) [27]. 3D-DNA scaffolding was performed by first mapping Hi-C reads to the contig assembly using Juicer v1.6

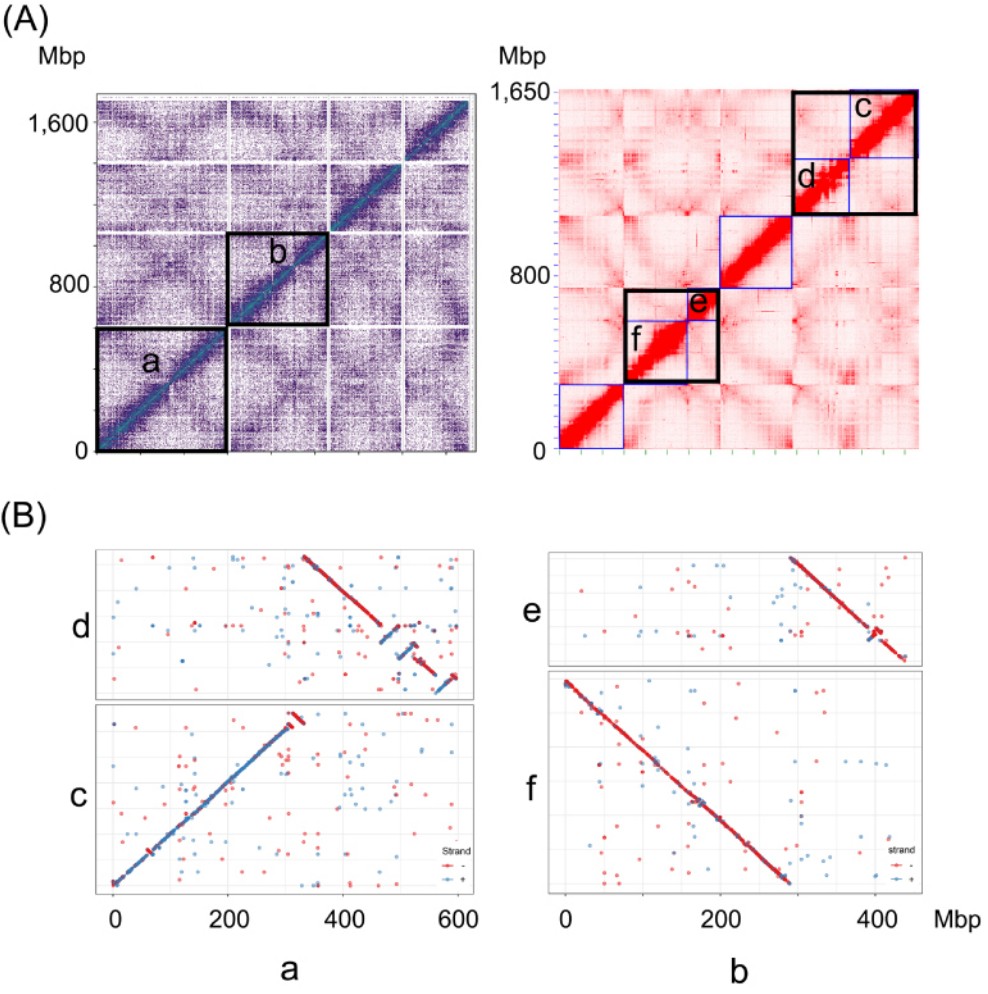

**Figure 3.** Resolving misjoin sites in *V. sativa* pseudo-chromosomes. (A) The left figure shows the interaction heatmap of four pseudo-chromosomes generated by the HiRise pipeline. Regions in black boxes show the potential misjoined pseudo-chromosomes indicated by weak interaction signals. After rescaffolding the genome using the 3D-DNA pipeline, mi-joins were confirmed and resulted in six pseudo-chromosomes (right figure, black boxes). (B) The collinearity between pseudo-chromosomes "a" to "c" and "d", and between pseudo-chromosomes "b" to "e" and "f" in (A) were confirmed by Mummer alignment.

(Juicer, RRID:SCR_017226) [28], which then generated 304,484,352 uniquely mapped pair-end reads and of which 51.1% (155,477,211) of the uniquely mapped reads were identified as valid Hi-C contacts. The 3D-DNA v180114 pipeline was integrated to anchor contigs to pseudo-chromosomes based on valid Hi-C contacts. The output file was used to generate a Hi-C heatmap for manual inspection using Juicebox Assembly Tools v1.11.08 (Juicebox, RRID:SCR_021172). This revealed six high-quality pseudo-chromosomes (Figure 3A).

We compared the HiRise and 3D-DNA assembled pseudo-chromosomes by performing a whole genome alignment with Mummer v4.0.0 (MUMmer, RRID:SCR_018171) [29]. The alignment showed a strong synteny between the HiRise and 3D-DNA pseudo-chromosomes (Figure 3B). However, the two longest HiRise pseudo-chromosomes aligned to four 3D-DNA pseudo-chromosomes suggesting two HiRise pseudo-chromosomes were misjoined



**Table 3.** The length of *V. sativa* pseudo-chromosomes.

| Pseudo-chromosome | Length (bp) |
|---|---|
| 1 | 324,818,257 |
| 2 | 324,640,943 |
| 3 | 290,752,327 |
| 4 | 290,123,409 |
| 5 | 272,590,232 |
| 6 | 148,681,034 |
| Total | 1,651,606,202 |

**Table 4.** Overview of *Vicia sativa* genome assembly.

| Feature | Value |
|---|---|
| Total length (bp) | 1,653,553,227 |
| No. of contigs | 9,990 |
| Contig N50 length (bp) | 684,593 |
| Scaffold N50 length (bp) | 290,126,875 |
| GC content (%) | 35.6 |
| Predicted protein-coding genes | 53,218 |
| Predicted noncoding genes | 3,966 |
| Content of repetitive sequences (%) | 83.92 |

(Figure 3B). The putative misjoined HiRise pseudo-chromosomes also coincided with low Hi-C interactions (Figure 3A).

To further support that these two HiRise pseudo-chromosomes were misjoined, we compared the synteny of the HiRise and 3D-DNA pseudo-chromosomes to the high-quality *V. faba* genetic linkage map [30] as no genetic linkage map is available for *V. sativa*. When we compared the order of 1536 sequenced *V. faba* DNA markers to their homologous regions in our HiRise and 3D-DNA pseudo-chromosomes, we observed a clear synteny between *V. faba* and *V. sativa*. However, two out of four of the HiRise pseudo-chromosomes appeared to be misjoined, for example, the markers on HiRise pseudo-chromosomes one, mapped to two *V. faba* linkage groups (Figure 4). After combining the karyotype, Hi-C interaction and synteny data to *V. faba*, we concluded the 3D-DNA assembly was most likely correct and subsequently used this assembly in further analysis. Finally, we used purge_dup pipeline v1.2.5 (purge dups, RRID:SCR_021173) [31] to remove low coverage scaffolds, partial overlaps and haplotigs. The final version of the genome assembly contains six pseudo-chromosomes (Table 3), in which a total of 1.65 Gbp contigs are anchored to these pseudo-chromosomes (Figure 5), and remain 10 unassigned contigs (overall 334,511 bp length). The overall genome size is 1,653,553,227 bp, with a GC content of 35.6% (Table 4).

## DATA VALIDATION AND QUALITY CONTROL

Three approaches were used to assess the quality of the final version of our genome assembly. First, the WGS short-read data was mapped to this final assembly. A very high proportion (99.7%) was mapped (Table 5). Second, the genome completeness was assessed by using BUSCO v5.2.2 referencing fabales_odb10 gene sets. Overall, BUSCO identified 97.8% complete genes (of which 8.9% were duplicated), 0.3% fragmented genes, and 1.9% missing genes out of 5366 markers in the gene sets. Finally, the LTR Assembly Index (LAI) of 12.96 was calculated by feeding the result of LTRharvest v1.6.2 (LTRharvest,

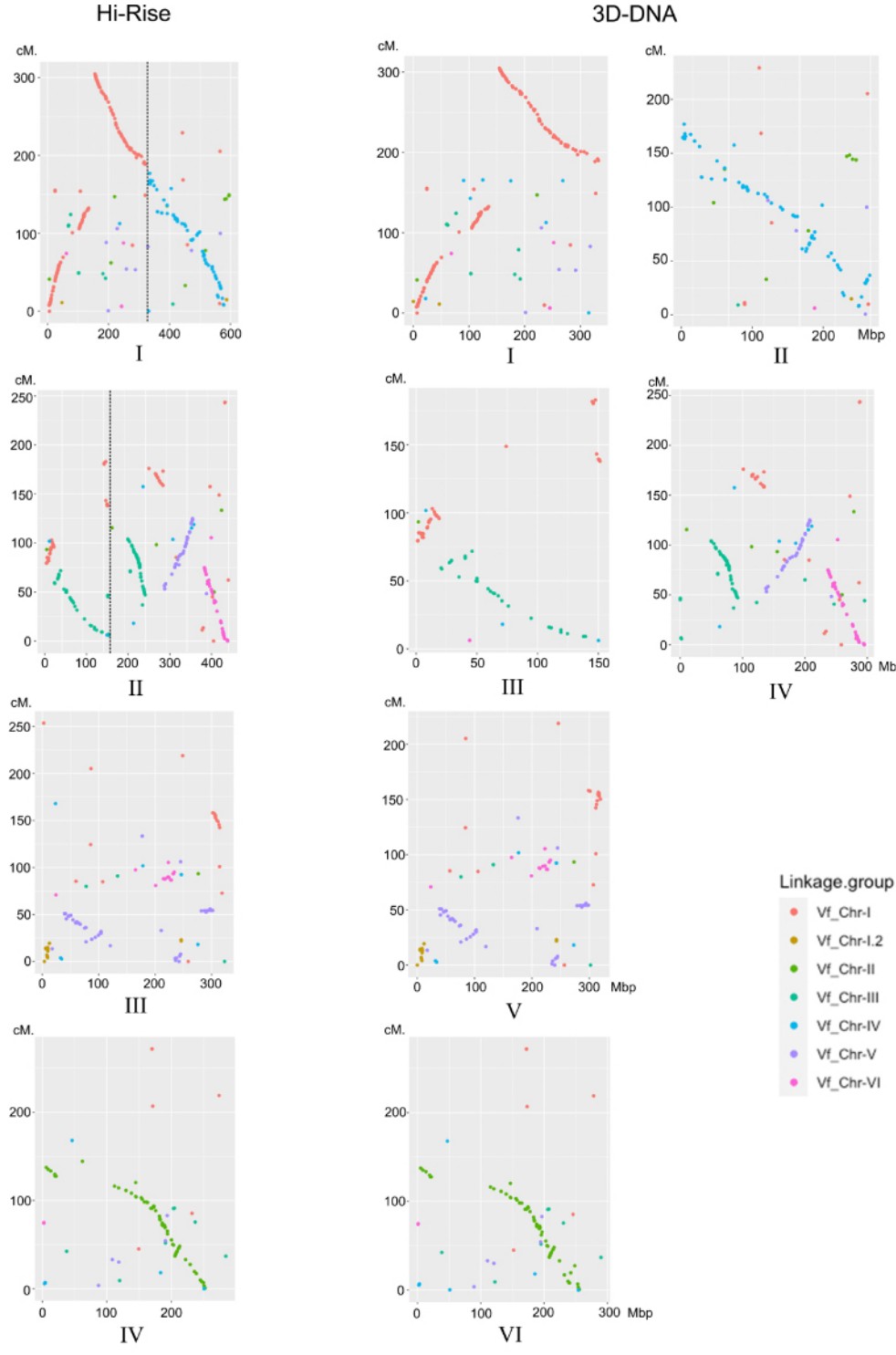

**Figure 4.** Comparison of HiRise and 3D-DNA-assembled pseudo-chromosomes to the *Vicia faba* genetic linkage map. Left, comparison of the four HiRise pseudo-chromosomes to the genetic linkage map; right, comparison of the six 3D-DNA pseudo-chromosomes to the linkage map. The *x* axes present the coordinates of the pseudo-chromosomes, the *y* axis presents the cumulative distance on the *V. faba* linkage map. Each color corresponds to a linkage group on the *V. faba* linkage map. 3D-DNA pseudo-chromosomes I and II are labelled as "a" and "b" in Figure 3, respectively. HiRise pseudo-chromosomes I, II, III and IV are labelled as "c", "d", "e" and "f" in Figure 3, respectively.

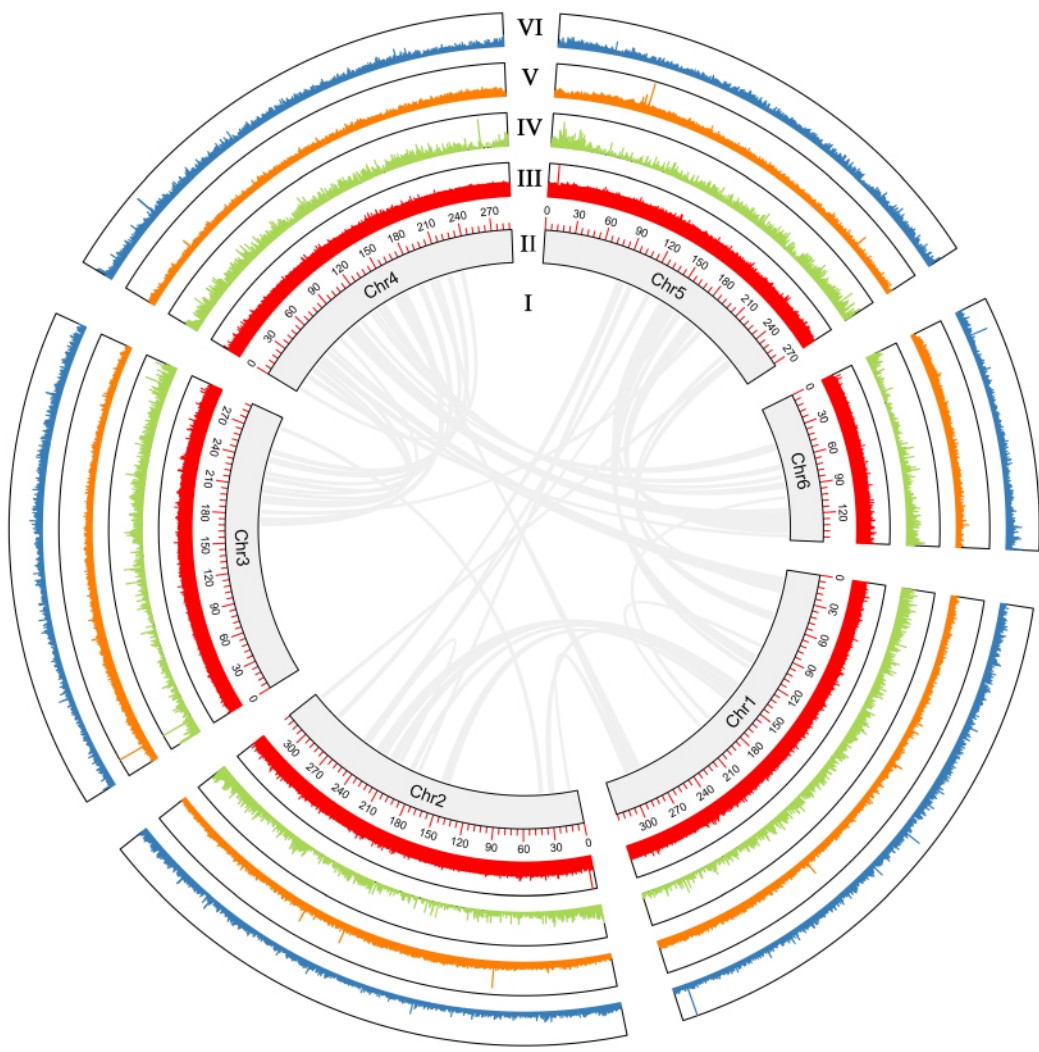

**Figure 5.** Circos plot showing the characterization of the *Vicia sativa* genome assembly. (I) Syntenic regions within the *V. sativa* genome based on homology searches using MCscan in Jcvi (MCScan, RRID:SCR_017650) [75] requiring ≥10 genes per block (links). (II) Pseudo-chromosome length in Mbp. (III) GC content in non-overlapping 10 Kbp windows (histograms). (IV) Gene density in non-overlapping 10-Kbp windows (histograms). (V) LTR-transposable element density in non-overlapping 10-Kbp windows. (VI) Mutator TIR transposon density in non-overlapping 10-Kbp windows (histograms). Percentage of GC content, gene density, and transposable element density were calculated relative to the highest value present in the genome. Chr = pseudo-chromosome.

RRID:SCR_018970) [32] and LTR_FINDER_parallel v1.2 [33] into LTR_retriever v2.9.0 (LTR_retriever, RRID:SCR_017623) [34], suggest that the genome reached a reference quality.

## Genome annotation

To annotate the *V. sativa* genome assembly, we masked repeat regions of the genome, then mapped the RNA-seq data to the masked genome and performed gene prediction. First, the repeat families found in the *V. sativa* genome were identified *de novo* and classified using the software package EDTA v1.9.6 [35] with the sensitive model setting. EDTA integrated multiple programs, including LTR_FINDER (LTR_Finder, RRID:SCR_015247) [36] and

**Table 5.** Mapping results of Illumina paired-end reads with short insert sizes.

| Parameters | Percentage (%) |
|---|---|
| Reads mapping rate | 99.7 |
| Genome coverage | 84.1 |
| Coverage at least 5× | 81.9 |
| Coverage at least 10× | 78.3 |
| Coverage at least 20× | 76.7 |

**Table 6.** Prediction of transposable element percentage in the *Vicia sativa* genome.

| Number of elements | | Number of elements | Length of occupied (bp) | % of genome |
|---|---|---|---|---|
| Retroelements | | 1,361,823 | 1,064,507,557 | 64.4 |
| | LINEs | 5,620 | 2,743,407 | 0.2 |
| | LTR elements | 1,356,203 | 1,061,764,150 | 64.2 |
| DNA transposons | | 704,467 | 242,003,507 | 14.6 |
| | Mutator TIR transposon | 209,091 | 116,510,919 | 7.0 |
| | hobo-Activator | 88 | 34,340 | 0.0 |
| | Tourist/Harbinger | 318 | 212,845 | 0.01 |
| Unclassified | | 319,392 | 69,154,926 | 4.2 |
| Simple repeats | | 174,030 | 10,230,793 | 0.6 |
| Low complexity | | 29,826 | 1,557,616 | 0.1 |
| Total | | 2,589,538 | 1,387,454,399 | 83.9 |

**Table 7.** Summary of gene predictions.

| Gene set | Number of genes | CDS + intron length (avg.) | CDS length (avg.) | Exon length (avg.) | Intron length (avg.) | Exons per gene (avg.) |
|---|---|---|---|---|---|---|
| Braker | 53,218 | 2267.11 | 956.97 | 223.43 | 415.13 | 4.42 |

RepeatModeler (RepeatModeler, RRID:SCR_015027), which generated a non-redundant transposable element (TE) library used to annotate the TE regions on the genome. The TE library generated from EDTA was also used as an input to RepeatMasker v4.1.2 (RepeatMasker, RRID:SCR_012954) to identify and perform "hard-masking" and "soft-masking" for the repetitive region on the genome. A total of 83.9% of the genome was masked, and 64.4% of the genome was detected as LTR elements (Table 6).

After genome masking, a combination of *ab initio* prediction and transcript evidence from the RNA-seq was used for gene prediction. Briefly, each RNA-seq data set was trimmed for low quality bases using TrimGalore v0.4.2, and mapped to the hard-masked-genome by using STAR v2.7.9 (STAR, RRID:SCR_004463) [37] to generate BAM files. Then the soft-masked genome and the BAM files generated from STAR were used for gene prediction using BRAKER v2.1.6 (BRAKER, RRID:SCR_018964) [38]. A total of 53,218 predicted protein-coding-genes were identified (Table 7). To assess the completeness of these protein-coding-genes, BUSCO v5.1.3 with fabales_odb10 gene sets were used which then identified 5127 (95.6%) complete, 395 (7.4%) duplicated, 70 fragmented (1.3%) and 169 missing (3.1%) orthologs.

Putative functions of the predicted protein-coding-genes were characterized by comparing the predicted proteins against the SwissProt and National Center for Biotechnology Information (NCBI) non-redundant database using Diamond v2.0.11 (DIAMOND, RRID:SCR_016071) [39] with e-value cut-off of $1 \times 10^{-5}$. Protein motifs and domains were annotated by comparing the predicted proteins against the InterPro

**Table 8.** Number of genes with homologs or functional classifications based on different databases.

| Database | | Annotated number | Annotated percentage (%) |
|---|---|---|---|
| NCBI-NR | | 44,400 | 83.4 |
| Swiss-Prot | | 31,071 | 58.4 |
| InterPro | All | 43,549 | 81.8 |
| | Pfam | 30,264 | 56.9 |
| | GO | 8,983 | 16.9 |
| Eggnog | Pfam | 34,527 | 64.9 |
| | KEGG_pathway | 10,777 | 20.3 |
| | KEGG_ko | 16,898 | 31.8 |
| | GO | 17,987 | 33.8 |
| Annotated | | 47,580 | 89.4 |
| Total | | 53,218 | — |

**Table 9.** Types of non-coding RNA detected from the *Vicia sativa* genome.

| Type | | Copy number | Average length (bp) | Total length (bp) | % of genome |
|---|---|---|---|---|---|
| miRNA | | 158 | 111.3 | 17,579 | 0.001 |
| tRNA | | 1382 | 73.7 | 101,891 | 0.006 |
| rRNA | rRNA | 649 | 440.1 | 285,638 | 0.017 |
| | 18S | 32 | 1763.5 | 56,431 | 0.003 |
| | 28S | 39 | 4249.9 | 165,745 | 0.010 |
| | 5S | 578 | 109.8 | 63,462 | 0.003 |
| snRNA | snRNA | 1777 | 107.5 | 191,047 | 0.011 |
| | CD-box | 1551 | 102.4 | 158,835 | 0.010 |
| | HACA-box | 69 | 126.7 | 8,740 | 0.001 |
| | splicing | 157 | 149.5 | 23,472 | 0.001 |

database using Interproscan v5.52-86.0 (InterProScan, RRID:SCR_005829) [40]. The predicted proteins were also assigned with Gene Ontology (GO) terms corresponding to the InterPro entries using Interproscan v5.52-86.0. In addition, we compared the predicted proteins against the EggNOG database v5.0 (eggnog, RRID:SCR_002456) [41] using eggNOG-mapper v2.1.4-2 (eggNOG-mapper, RRID:SCR_021165) [42] and assigned them with Kyoto Encyclopedia of Genes and Genomes (KEGG) pathways and KEGG orthologous groups (KO). As a result, we were able to annotate 47,580 (89.4%) predicted protein-coding genes with at least one function term (Table 8).

In addition, we also identified and annotated non-coding RNA in the *V. sativa* genome. tRNA was identified using tRNAscan-SE v2.07 (tRNAscan-SE, RRID:SCR_010835) [43], rRNA was identified using Rnammer v1.2 (RNAmmer, RRID:SCR_017075) [44] and other types of non-coding RNA were identified by using Infernal v1.1.4 (Infernal, RRID:SCR_011809) [45] based on the Rfam database (Rfam, RRID:SCR_007891) [46]. Overall, 3966 of noncoding genes were annotated, including 158 miRNA, 649 rRNA and 1777 snRNA (Table 9).

## Phylogenetic tree construction and divergence time estimation

We identified the orthogroups, phylogenetic positions and divergence times between *V. sativa* and 11 other plant species. The source of the protein-coding sequences used in our analysis are listed in Table 10. First, protein sequences of *V. sativa*, *Pisum sativum*, *Medicago truncatula*, *Trifolium pratense*, *Phaseolus vulgaris*, *P. lunatus*, *Vigna unguiculata*, *Chamaecrista fasciculata*, *Faidherbia albida*, *Cercis canadensis*, *Carya illinoinensis*, and *Arabidopsis thaliana* [47–54] were clustered into orthogroups using Orthofinder v2.5.4

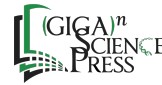

**Table 10.** A list of species and their associated sequencing data used in our study.

| Species | Abbreviation name | Source of data | Data version |
|---|---|---|---|
| *Vicia sativa* | *V. sat* | This project | |
| *Pisum sativum* | *P. sat* | URGI | V1a |
| *Medicago truncatula* | *M. tru* | INRA | MtA17 r5 |
| *Trifolium pratense* | *T. pra* | Phytozome | v2 |
| *Phaseolus vulgaris* | *P. vul* | Phytozome | v2.1 |
| *Phaseolus lunatus* | *P. lun* | Phytozome | v1 |
| *Vigna unguiculata* | *V. ung* | Phytozome | v1.2 |
| *Chamaecrista fasciculata* | *C. fas* | GigaDB | v1 |
| *Faidherbia albida* | *F. alb* | GigaDB | N/A |
| *Cercis canadensis* | *C. can* | GigaDB | v1 |
| *Carya illinoinensis* | *C. ill* | Phytozome | v1.1 |
| *Arabidopsis thaliana* | *A. tha* | Phytozome | TAIR10 |

**Table 11.** Summary of genes and orthogroups for species used in this study.

| Species | Number of genes | Number of orthogroups | Number of genes in orthogroups | Number of species-specific orthogroups | Number of genes in species-specific orthogroups | Single copy genes |
|---|---|---|---|---|---|---|
| *V. sat* | 53,218 | 19,096 | 48,028 | 1774 | 8,594 | 10,009 |
| *P. sat* | 57,835 | 19,012 | 51,576 | 2203 | 10,289 | 8,131 |
| *M. tru* | 44,618 | 18,528 | 38,693 | 909 | 3,180 | 10,755 |
| *T. pra* | 39,943 | 18,366 | 36,476 | 791 | 2,558 | 10,686 |
| *P. vul* | 27,433 | 16,521 | 26,884 | 47 | 137 | 10,660 |
| *P. lun* | 43,997 | 16,918 | 42,007 | 408 | 7,518 | 10,730 |
| *V. ung* | 31,948 | 16,741 | 30,176 | 336 | 1,463 | 10,297 |
| *C. fas* | 32,832 | 14,944 | 31,229 | 472 | 4,336 | 9,630 |
| *F. alb* | 28,979 | 15,695 | 26,573 | 450 | 1,666 | 9,883 |
| *C. can* | 34,023 | 16,165 | 32,407 | 694 | 3,767 | 12,289 |
| *C. ill* | 31,911 | 15,424 | 30,007 | 528 | 2,501 | 7,830 |
| *A. tha* | 27,416 | 14,171 | 24,887 | 870 | 4,286 | 8,851 |

(OrthoFinder, RRID:SCR_017118) [55] with default parameters. A total of 10,009 single-copy and 43,209 multi-copy genes were identified in the *V. sativa* annotation (Figure 6B, Table 11), forming 19,096 orthogroups (Figure 6A, Table 10). Comparing orthogroups amongst *V. sativa, P. sativum, M. truncatula, P. vulgaris, F. albida*, we identified 2309 orthogroups that are specific to *V. sativa* (Figure 6A). Orthofinder was further used to perform phylogenetic reconstruction with the multiple sequence alignment approach (using the -msa command). The generated species tree has a support value of one on all nodes (Figure 7), indicating the high reliability of the revealed phylogenetic relationships.

To estimate divergence times between *V. sativa* and other important legume species (Table 10), coding sequences of 64 randomly selected single copy orthologs (see Supplementary File 1 [57]) were aligned using MACSE v1.2 [58]. Low-quality regions of each alignment were trimmed using Trimal v1.4.1 (trimAl, RRID:SCR_017334) [59], resulting in high-quality alignments amounting to 139,956 bp. Individual alignments were then imported into Beast v2.6.3 (BEAST2, RRID:SCR_017307) [60] for phylogenetic dating. Substitution models were selected using BEAST Model Test [61] for each alignment and were allowed to coalesce using unlinked relaxed log-normal molecular clocks [62]. A calibrated Yule prior [63] was used to inform tree building and speciation with four node calibrations (Table 12). First, a log normal distribution of 89.3 MYA (5% quantile 97.9 million years ago [MYA], median 106 MYA, 95% quantile 121 MYA) [64] was used to inform the root





**Figure 6.** Evolution of the *V. sativa* genome. (A) A Venn diagram showing shared and unique orthologous gene families in *V. sativa* and four other legumes. (B) Predicted orthologous protein composition for *V. sativa* compared to *A. thaliana*, *C. illinoinensis* and nine legumes. (C) A phylogenetic tree shows the expansion and contraction of the gene families and the divergence time for species. (D) Ks plot shows the whole genome duplication event in *V. sativa*, *M. truncatula* and *P. vulgaris*. *V. sat*: *Vicia sativa*, *P. sat*: *Pisum sativum*, *M. tru*: *Medicago truncatula*, *T. pra*: *Trifolium pratense*, *P. vul*: *Phaseolus vulgaris*, *P. lun*: *Phaseolus lunatus*, *V. ung*: *Vigna unguiculata*, *C. fas*: *Chamaecrista fasciculata*, *F. alb*: *Faidherbia albida*, *C. can*: *Cercis canadensis*, *C. ill*: *Carya illinoinensis*, *A. tha*: *Arabidopsis thaliana*.

prior (Brassicaceae, Fabaceae split). Three fossil calibrations were then set using CladeAge [65]: (i) Fabaceae (Figure 8 red dot; 65.3 MYA) [66], (ii) Caesalpinioideae (Figure 8 blue dot; 58 MYA) [67, 68], and (iii) Papilionoideae (Figure 8 green dot; 55 MYA) [69]. Furthermore, a net diversification rate was set to 0.1–0.134 to construct a distribution around the literature value of 0.117 [70], turnover rate was set to 0.823–0.883 to construct a distribution around the literature value of 0.853 [70], and sampling rate of 0.000034–0.013 [71] was set to determine CladeAge prior distributions. Final chain length of the Markov Chain Monte Carlo (MCMC; 600 million) was determined through continuous examination of the log file using Tracer until proper mixing was observed. This allowed us to determine a robust estimate for the most common recent ancestor (MRCA) of *V. sativa* and *P. sativum* at 10.6 (95% Highest Posterior Density: 9.9–11.4) MYA (Figure 8). Gene family

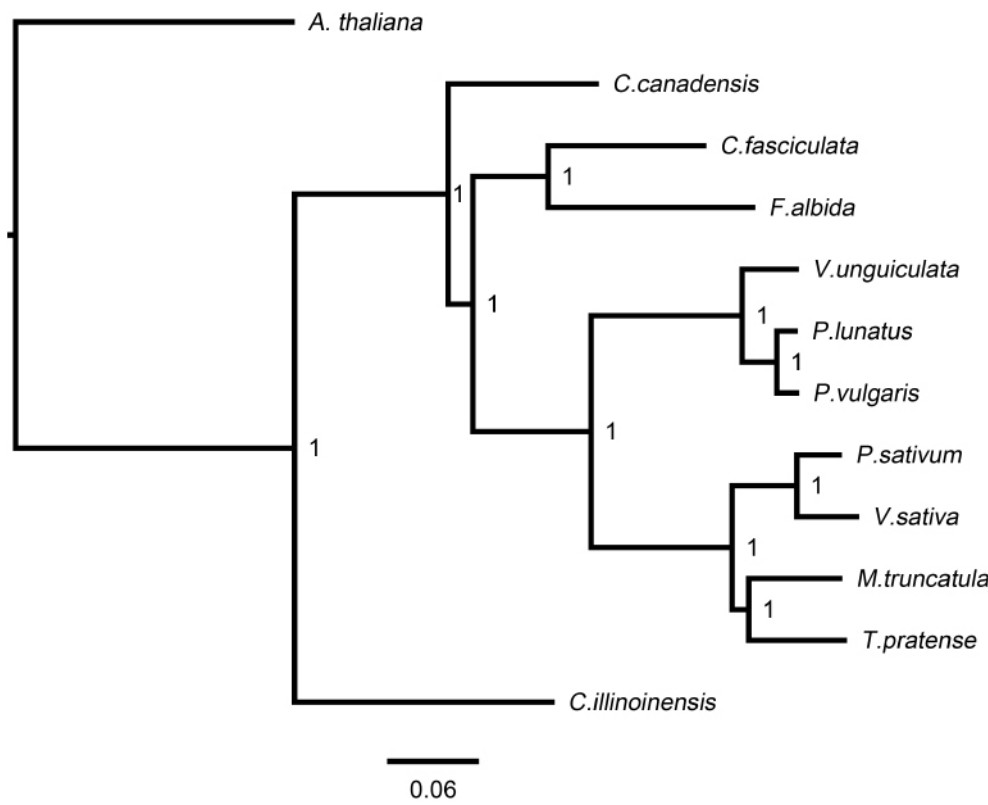

**Figure 7.** Species tree generated by Orthofinder using a multiple sequence alignment approach. Node label shows the Shimodaira–Hasegawa test supporting value [56].

| Table 12. | Fossil records used in divergence time analysis. | | |
|---|---|---|---|
| **Node** | **Definition** | **Fossil** | **Age (Ma)** |
| Yellow | SG Brassicales | Flowers of *Dressiantha bicarpellate*; USA | 89.3 |
| Red | SG Leguminosae | Seedpods and leaflets; USA | 65.3 |
| Blue | SG Caesalpinioideae | Bipinnate leaves; Colombia | 58 |
| Green | SG Papilionoideae | Flowers of *Barnebyanthus buchananensis*; USA | 55 |

expansion and contraction analysis using CAFE v4.2.1 (Computational Analysis of gene Family Evolution, RRID:SCR_018924) [72] with a single $\lambda$ revealed 5195 gene families that have undergone gene expansion (3024) or contraction (2171) since the MRCA of *V. sativa* and *A. thaliana* (Figure 6C).

To identify whole genome duplication events (WGD), WGDI v0.5.1 [73] was used to identify gene collinearity between *V. sativa*, *M. truncatula* and *P. vulgaris*. The $K_s$ (synonymous substitutions per synonymous site) value was calculated based on the identified collinearity gene to construct a frequency distribution map. The Ks distribution indicated that *V. sativa*, *M. truncatula* and *P. vulgaris* share the same ancestral WGD event. The estimated time of this WGD event (~58 MYA) [74] and the corresponding Ks value (~0.93, Figure 6D) reveal that the average mutation rate of *V. sativa* genome is 8.02 ×10$^{-9}$ per site per year.

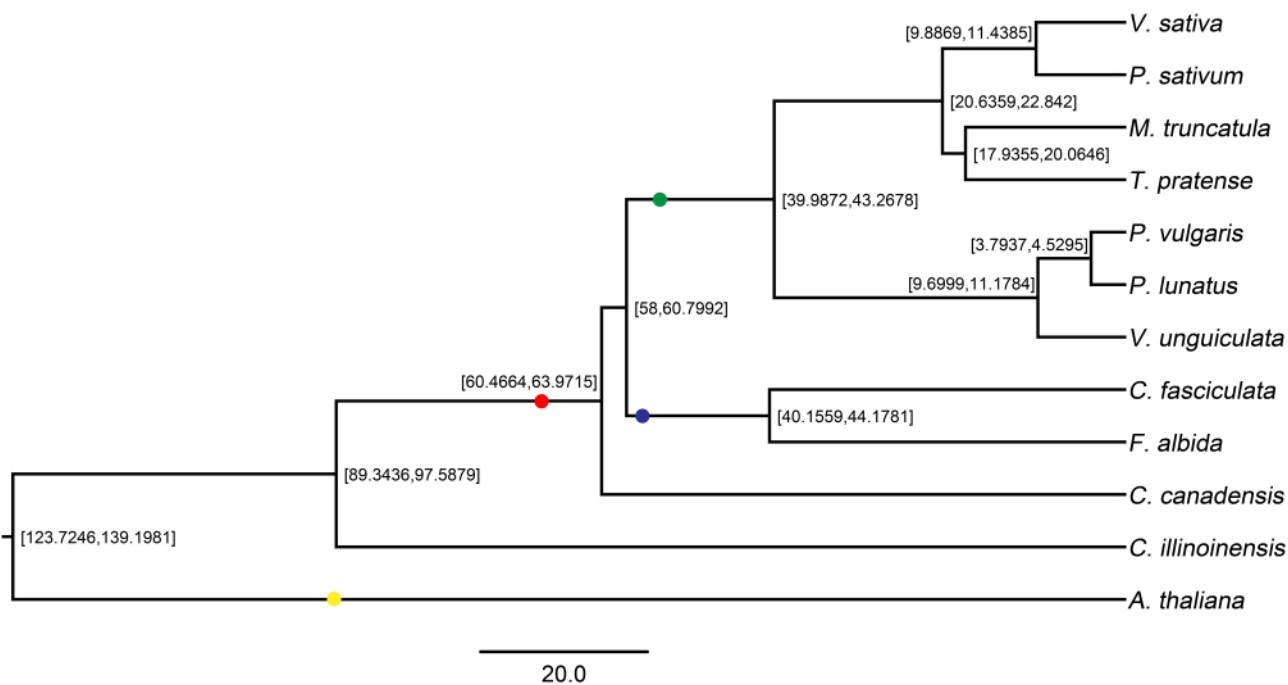

**Figure 8.** Divergence time estimation for *Vicia sativa* and other species. The node labels show the 95% Highest Posterior Density of species divergence time. Units shown on the scale bar are million years ago. Yellow, red, blue and green dots correspond to fossil calibration points.

## REUSE POTENTIAL

Understanding the genetic, epigenetic and epitranscriptomic basis of the evolutionary processes shaping drought tolerance, low nutrient requirements and adaption to broad habitats requires comparison of multiple legume genomes, preferably assembled at the chromosome level. In this study, we present a complete chromosome-level genome assembly for the legume *V. sativa* (common vetch) and provided a detailed genome annotation. There are >19,000 species of legumes, about 200 within the *Vicia* genus, and this genome will serve as an excellent reference for the assembly of other *Vicia* genomes. The *V. sativa* genome will also facilitate comparative analyses aimed at understanding the evolutionary origin and dynamics of legume specific gene families. Our new *V. sativa* genome will greatly benefit legume researchers and plant breeders who are interested in conventional as well as engineering crop improvement.

## DATA AVAILABILITY

Final assembly and original Nanopore assembly, as well as annotation files, Supplementary File 1, predicted transcript and protein sequences, and bioinformatics supporting information, were deposited in the database GigaDB [57]. Additionally, assembly, Illumina and Nanopore subreads, and transcriptome raw data are available at NCBI and can be accessed with BioProject PRJNA762450 and BioSample SAMN21393724. Illumina and Nanopore subreads can be obtained, with SRR16004114 and SRR16004115; and RNA-sequencing raw reads, as follows: SAMN21545804, SAMN21545805, SAMN21545806, SAMN21545807 and SAMN21545808. Additional data is available in the *GigaScience* GigaDB database [57].

## DECLARATIONS

### List of abbreviations

CHiCAGO: Capture Hi-C Analysis of Genomic Organisation; Gbp: gigabase pairs; Kbp: kilobase pairs; Mbp: megabase pairs; MCRA: most common recent ancestor; MYA: million years ago; NCBI: National Center for Biotechnology Information; TE: transposable element; WGS: whole genome sequencing, WGD: whole-genome duplication.

## ETHICAL APPROVAL

Not applicable.

## CONSENT FOR PUBLICATION

Not applicable.

## COMPETING INTERESTS

The authors declare that they have no competing interests.

## FUNDING

This work was funded by the Department of Industry, Science, Energy and Resources (grant number ACSRF 48187), the National Natural Science Foundation of China (grant number 31722055) and an Australia Research Council Future Fellowship (grant number FT130100525) awarded to IRS. HX, CW and VN were supported by University of Adelaide Research Training Scholarships (RTS) and University research support. VN was also supported by an AW Howard Memorial Trust Postgraduate Research fellowship.

## AUTHORS' CONTRIBUTIONS

HW conducted the genome assembly, genome analysis and wrote the manuscript. VN prepared DNA and RNA for sequencing and co-wrote the manuscript. CW assisted with the genome assembly. IRS conceived and managed the project, interpreted the data and drafted the figures. ZL interpreted the data. All authors read, edited, revised, and approved the manuscript final version.

## ACKNOWLEDGEMENTS

We are grateful to the high-performance computing infrastructure provided by the University of Adelaide. We thank the Australian National Vetch Breeding program for gifting the *V. sativa* seed.

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
