## [Reviewer Report]

Comments on revised manuscriptThe revised manuscript and response are satisfactory. The additional analyses that the authors have performed are correctly structured. The data presented is clear. In my opinion, I recommend accepting this manuscript.

---

## [Editor Report]

Comments to the AuthorPlease check the GigaDB DOI if there are any changes to the metadata required.

---

## [Reviewer Report]

Reviewer name and names of any other individual's who aided in reviewer Jonathan KreplakDo you understand and agree to our policy of having open and named reviews, and having your review included with the published papers. (If no, please inform the editor that you cannot review this manuscript.)YesIs the language of sufficient quality?YesPlease add additional comments on language quality to clarify if needed
Are all data available and do they match the descriptions in the paper? YesAdditional CommentsAre the data and metadata consistent with relevant minimum information or reporting standards? See GigaDB checklists for examples <a href="http://gigadb.org/site/guide" target="_blank">http://gigadb.org/site/guide</a>YesAdditional CommentsIs the data acquisition clear, complete and methodologically sound?YesAdditional Comments

Is there sufficient detail in the methods and data-processing steps to allow reproduction?NoAdditional CommentsFor "Phylogenetic tree construction and divergence time estimation", 64 single copy orthologs are selected, they should be included in a supplementary table to be able to fully reproduct the analysis. Also, Supplementary table S9 should be related to fossil calibrations but show the length of chromosome. Is there sufficient data validation and statistical analyses of data quality? YesAdditional CommentsIs the validation suitable for this type of data?YesAdditional CommentsIs there sufficient information for others to reuse this dataset or integrate it with other data?YesAdditional CommentsAny Additional Overall Comments to the AuthorThis work is a state of the art assembly for Vicia Sativa and will be useful to understand how the Vicia genome size has expanded. The scientific part is lacking a few elements that could boost the manuscript. On Vicia Faba, satellite associated to centromeres are well-studied. It 's interesting to check and could confirm that the assembly is robust. Functional annotation seems solid but I was surprised by the low number of genes annotated with a GO using interpro. (Table S6) I would have expected an higher number. Is there an error ? eggNOG-mapper is also giving a GO assignation but the percentage isn't reported in the table. Are they similar ? 
More disappointing for me, orthologs analysis between species is well done and sufficient but missing a few sequenced legumes genome like P.sativum. Also, M.truncatula have a newer version that the one used. A ressource like legum federation (https://www.legumefederation.org/ ) could have been helpful to select the most adequate genome for this analysis. 

Figures : 
One of the main figure (fig2) doesn't seems right. LTR (V) represent 60% of the genome sequence but appear as abundant as TIR (VI) which are a subclass of Transposons (less than 16% of the genome). This is due to the fact that scale is different for each track. A common scale (like a density) must be used . Also, you should add TIR percentage in table S3. For me, this figure isn't informative enough and must be rework. 

Fig3 (D) green line legend is false and should be changed

RecommendationMinor Revision

---

## [Reviewer Report]

Reviewer name and names of any other individual's who aided in reviewer Jianbo JianDo you understand and agree to our policy of having open and named reviews, and having your review included with the published papers. (If no, please inform the editor that you cannot review this manuscript.)YesIs the language of sufficient quality?YesPlease add additional comments on language quality to clarify if needed
Are all data available and do they match the descriptions in the paper? YesAdditional CommentsAre the data and metadata consistent with relevant minimum information or reporting standards? See GigaDB checklists for examples <a href="http://gigadb.org/site/guide" target="_blank">http://gigadb.org/site/guide</a>YesAdditional CommentsIs the data acquisition clear, complete and methodologically sound?YesAdditional CommentsIs there sufficient detail in the methods and data-processing steps to allow reproduction?YesAdditional CommentsIs there sufficient data validation and statistical analyses of data quality? YesAdditional CommentsIs the validation suitable for this type of data?YesAdditional CommentsIs there sufficient information for others to reuse this dataset or integrate it with other data?YesAdditional CommentsAny Additional Overall Comments to the AuthorIn this manuscript, Xi et al reported a chromosome-level genome of the common vetch (Vicia sativa) with integration of Oxford Nanopore sequencing, Illumina sequencing, CHiCAGO and Hi-C. Then, the gene annotation and evolution were performed based on the reference genomes. These genomic resources are valuable for evolution research, genetic diversity and genomic breeding. I think this manuscript is suitable published in Gigabyte. Some minor comments and suggestions as following:

1) The Line Number is missed in this manuscript, which make the detailed comments is not inconvenient.
2) Page 6, “resequenced short-reads” should be “De novo sequencing” or “sequencing”.
3) For the 1.93 Gb assembled genome size, it is a little larger than that of estimated by the flow cytometry (1.77 Gb) and Genomescope (1.61 Gb). Maybe there are some duplicated sequences in this version of assembled genomes. Some redundancy removal software can deal with this question such as Haplotigs, Purge_dups and so on.
4) For the evaluation of genome, LTR Assembly Index (LAI) was suggested for the quality assessment. 
5) In Table S2, the mapping rate is very well but the genome coverage is just 76% which looks a little low. What’s the reason?
6) In Table S4, the gene set was combined by August. However, in methods, the annotation software is BRAKER v2.1.6.
RecommendationMinor Revision